# Peri-Implant Bone Loss and Overload: A Systematic Review Focusing on Occlusal Analysis through Digital and Analogic Methods

**DOI:** 10.3390/jcm11164812

**Published:** 2022-08-17

**Authors:** Adolfo Di Fiore, Mattia Montagner, Stefano Sivolella, Edoardo Stellini, Burak Yilmaz, Giulia Brunello

**Affiliations:** 1Department of Neurosciences, School of Dentistry, University of Padova, 35128 Padova, Italy; 2Private Practice, 35121 Padova, Italy; 3Department of Reconstructive Dentistry and Gerodontology, School of Dental Medicine, University of Bern, 3012 Bern, Switzerland; 4Department of Restorative, Preventive and Pediatric Dentistry, School of Dental Medicine, University of Bern, 3012 Bern, Switzerland; 5Division of Restorative and Prosthetic Dentistry, The Ohio State University, Columbus, OH 43210, USA; 6Department of Oral Surgery, University Hospital Düsseldorf, 40225 Düsseldorf, Germany

**Keywords:** dental implant, occlusion, overloading, complications, implant-supported restorations, marginal bone loss

## Abstract

The present review aimed to assess the possible relationship between occlusal overload and peri-implant bone loss. In accordance with the PRISMA guidelines, the MEDLINE, Scopus, and Cochrane databases were searched from January 1985 up to and including December 2021. The search strategy applied was: (dental OR oral) AND implants AND (overload OR excessive load OR occlusal wear) AND (bone loss OR peri-implantitis OR failure). Clinical studies that reported quantitative analysis of occlusal loads through digital contacts and/or occlusal wear were included. The studies were screened for eligibility by two independent reviewers. The quality of the included studies was assessed using the Risk of Bias in Non-randomized Studies of Interventions (ROBINS-I) tool. In total, 492 studies were identified in the search during the initial screening. Of those, 84 were subjected to full-text evaluation, and 7 fulfilled the inclusion criteria (4 cohort studies, 2 cross-sectional, and 1 case-control). Only one study used a digital device to assess excessive occlusal forces. Four out of seven studies reported a positive correlation between the overload and the crestal bone loss. All of the included studies had moderate to serious overall risk of bias, according to the ROBINS-I tool. In conclusion, the reported data relating the occlusal analysis to the peri-implant bone level seem to reveal an association, which must be further investigated using new digital tools that can help to standardize the methodology.

## 1. Introduction

Implant dentistry represents a safe and predictable treatment modality to rehabilitate both complete and partially edentulous patients [1]. The number of dental implants fitted every year is increasing; in the US, their prevalence rose from 0.7% in 1999 to 5.7% in 2015, with a projection of 23% in 2026 [1]. This tendency can be ascribed to an increase in oral-health-related quality of life [2,3]. In a recent systematic review, which included longitudinal studies with a follow-up of at least 10 years for a total of 7711 implants, a cumulative mean survival rate of 94.6% (SD 5.97%) was reported, with variation from 73.4% to 100% [4]. 

Several factors are reported to be associated with crestal bone loss (CBL), including bacterial colonization and the presence of a micro-gap between abutment and implant [5,6]. Contradictory findings on the role of occlusal overload on peri-implant bone loss have been reported, with limited evidence supporting the cause-and-effect relationship [7]. Overload is generally considered to be an excessive occlusal load on the implant-supported fixed dental prosthesis (FDP), leading to high stress on the peri-implant bone tissue [8,9,10]. An imbalance in the occlusal load may generate stress at the bone–implant coronal first contact point [11], which might increase the incidence of CBL [12]. 

The potential detrimental effect of overload in implant therapy was first observed by Adell [13] in 1981. Quirynen et al. [14] examined the effect of overload, finding an excessive CBL in the first year of load in patients who were rehabilitated using implant-supported prosthesis in both jaws and who presented a lack of anterior guidance or parafunctional activity. Naert et al. [15] also reported similar results under the same conditions. In more recent studies [16,17], which either analyze the characteristics of early and late implant failure or assess implant survival rates in bone of different qualities, implants failed due to more marked occlusal areas identified through articulating paper. However, many reviews reported a lack of evidence regarding the positive correlation between the overload and the CBL [7,18,19,20,21]. These heterogenous results may be attributable to the different methods of analyzing the overload. Clinical studies have assessed the presence of overload in relation to the length of cantilever, bruxism, tooth clenching, or the presence of an implant-supported prosthesis as the antagonist [18,19,20,21]; however, the results are difficult to compare and repeat. With the development of digital technology, some devices have been introduced in the dental market to assess occlusal force, but few are discussed in the articles published in the literature [22,23,24]. However, almost all of the articles concluded that using digital technology allows for more accurate constructions and the more precise balancing of occlusal relationships [22,23,24]. 

Contradictory results were found in animal studies. Some authors reported increased marginal bone loss [25], a loss of osseointegration [26], or crater-like bone resorption [27] in the presence of overload. However, excessive loading did not result in any difference in histologically assessed peri-implant bone loss, either in healthy implants nor in implants affected by ligature-induced peri-implantitis in primates [28]. In another study in dogs, no difference was reported in terms of the loss of osseointegration or marginal bone loss between non-loaded implants and implants subjected to excessive occlusal load after eight months [29]. In an animal study where a lateral load was applied to implants for 24 weeks, a structural adaptation of the peri-implant bone was histologically observed in the test implants compared to unloaded controls, in terms of higher bone density and mineralized bone-to-implant contact [30].

Overload potentially plays a role in the behavior of peri-implant bone; however, its role in the onset and progression of bone loss is still unclear. Therefore, a systematic review is needed to give a clear idea of the problem. This uncertainty has also been attributed to the difficulties in measuring the magnitude and the direction of forces in clinical studies [7]. The heterogeneity of the data regarding this subject perpetuates doubts among researchers and clinicians. Furthermore, to the best of the authors’ knowledge, there is no previous review on the topic focusing only on the utilization of repeatable and quantifiable overload assessment methods. Therefore, due to the lack of clear results, this systematic review aimed to assess the possible relationship between overload, assessed through digital occlusal analysis and/or occlusal wear, and crestal bone loss. 

## 2. Materials and Methods

To investigate the possible correlation between overload and crestal bone loss (CBL), an extensive search was conducted to identify scientific studies focused on the problem. This systematic review was conducted in accordance with the Preferred Reporting Items for Systematic Reviews (PRISMA) 2020 statement [31]. The protocol for this review was registered with the international prospective register of systematic reviews (PROSPERO) with registration no. CRD42021250518. The clinical question was formulated using the PICO strategy (population: patients with an osseointegrated implant; intervention: implant occlusal overload; comparison: absence of implant occlusal overloading; outcome: crestal bone loss). The PICO question was structured as follows: “Does implant occlusal overloading influence crestal bone loss around osseointegrated dental implants?”.

### 2.1. Search Strategy

A broad electronic search for relevant publications, published from 1 January 1985 to 31 December 2021, was performed across MEDLINE (via PubMed), the Cochrane Central Register of Controlled Trials (CENTRAL), and Scopus. The electronic search strategy applied was: (dental or oral) AND implants AND (overload OR excessive load OR occlusal wear) AND (bone loss OR peri-implantitis OR failure). No language restrictions were applied. To identify other eligible studies, a manual search based on the reference lists of the most relevant systematic reviews on the topic, and of all the articles retrieved from the electronic databases, was conducted. 

### 2.2. Inclusion and Exclusion Criteria

The following criteria had to be met for inclusion: clinical human studies, including randomized controlled trials (RCTs), cohort prospective or retrospective studies, case-control studies, and cross-sectional studies, which evaluated bone loss around osseointegrated dental implants subjected to overload (assessed through digital occlusal analysis and/or occlusal wear) with a follow-up of one or more years after prosthetic loading [17,19]. The following were excluded: animal studies, case reports, case series, guidelines, reviews, and in silico (3D element finite analysis) and in vitro studies. Clinical studies assessing the presence of overload by the length of cantilever, bruxism, tooth clenching, or the presence of an implant-supported prosthesis as an antagonist were also excluded due to the impossibility of comparing the data.

### 2.3. Study Selection and Data Extraction

The published articles were first screened by one reviewer (M.M.), by title and abstract. In the second step, the full texts of the selected articles were evaluated by two independent reviewers (A.D.F. and M.M.). The agreement between the two reviewers was assessed by means of the Cohen’s Kappa coefficient. Disagreements were resolved by discussion between the authors. The following data were extracted: title, authors, year of publication, journal in which the research was published, study design, number of patients and implants, patient characteristics, implant characteristics, type of prosthesis, follow-up, assessment methods (occlusal wear assessment), and main results. To simplify the terminology, all the terms used in the literature to identify the radiographic changes of peri-implant bone over time (e.g., crestal bone level, marginal bone level, crestal bone loss, marginal bone loss) were combined under the acronym CBL (crestal bone loss) and used as synonyms throughout this systematic review. Data were sought to find a difference in mean CBL (in mm) between overloaded and non-overloaded implants. Authors were contacted in order to acquire missing information, when necessary. 

### 2.4. Quality Assessment

The quality of each included study was individually assessed. In accordance with the Cochrane Collaboration guidelines, the Risk of Bias in Non-randomized Studies of Interventions (ROBINS-I) tool was utilized [32]. Using this tool, seven domains (i.e., confounding, selection, classification of interventions, deviations from the intended intervention, missing data, measurement of outcomes, and reporting results) for each included study were classified at “low”, “moderate”, “serious”, or “critical” risk of bias. Then, an overall score was given, judging the study at “low risk of bias” when it was assessed “low” in all domains, at “moderate risk of bias” when it was assessed “low” or “moderate” in all domains, at “serious risk of bias” when it was assessed “serious” in at least one domain, or at “critical risk of bias” when it was assessed “critical” in at least one domain.

## 3. Results

The flow diagram of the search results is presented in Figure 1. 

The electronic search produced a total of 469 potentially relevant publications. Then, 23 additional records were found through a manual search, yielding a total of 492 studies. After the removal of duplicated studies, 472 records were obtained, of which 388 were excluded after title and abstract screening. After full-text evaluation of 84 records, only seven articles fulfilled the inclusion criteria and were included for qualitative analysis. The main reasons for exclusion were: animal research (*n* = 21) [25,26,27,28,29,30,33,34,35,36,37,38,39,40,41,42,43,44,45,46], reviews (*n* = 26) [6,7,9,12,14,18,19,20,47,48,49,50,51,52,53,54,55,56,57,58,59,60,61,62,63,64], in silico (*n* = 11) [65,66,67,68,69,70,71,72,73,74,75], in vitro (*n* = 3) [76,77,78], guidelines (*n* = 3) [21,79,80], case reports (*n* = 6) [10,81,82,83,84,85], lack of occlusal assessment (*n* = 1) [86], overload assessment method (*n* = 6), i.e., by maximum bite force [87], or bruxism habits [88,89,90], or length of cantilever [91], or type of antagonist [15]. 

The kappa values for inter-reviewer agreement for full-text selection was 0.89, indicating high agreement between the reviewers. Of the seven included articles, four were cohort studies [92,93,94,95], two were cross-sectional studies [96,97], and one was a case-control study [98]. No RCT was found to be eligible. Details of the included studies are reported in Table 1.

Four studies [92,96,97,98] found a positive correlation between overload and CBL, while in the other three [93,94,95] no correlation was found. In two studies [96,98], a correlation was found between overload and peri-implantitis, which includes the radiographic detection of peri-implant bone loss as assessed according to the provided definition. Specifically, the case control study of Canullo et al. [98] identified the presence of overload with OR [95% CI] = 18.70 [5.5–63.2] (*p* < 0.001) as a predictor of peri-implantitis. Furthermore, Dalago et al. [96] found a positive relationship between peri-implantitis and prosthetic wear facets on crown and dentures in the univariate and in the multi-factor analysis (OR [95% CI] = 2.4 [1.2–4.8] *p* = 0.032). Kissa [97] reported higher probing depth and CBL in patients with facets on two or more posterior teeth. For Lindquist et al. [92], the length of the cantilever extensions and occlusal wear tended to be two factors implicated in increased CBL around the mesial implants. Indeed, the authors demonstrated a correlation between occlusal wear and CBL [92]. Conflicting results were reported in remaining three studies [93,94,95], which concluded that occlusal wear did not affect the annual vertical bone loss rate. The risk of bias in the seven studies included was assessed, and is summarized in Table 2.

All of the included studies had moderate to serious overall risk of bias. Specifically, all of the studies investigated possible confounding factors. Three studies [96,97,98] were deemed to have “moderate risk of bias” for the selection of participants, due to their retrospective design. All of the studies except one [98] presented a “moderate risk of bias” for classification of intervention, because the overload presence was identified by occlusal wear. Engel et al. [95] had a follow-up rate < 80% and no description of patients lost, hence it had a severe risk of bias in missing data. None of the four cohort studies [92,93,94,95] had an independent blind assessment of the outcomes. Only three studies [96,97,98] reported a complete data set of the results.

## 4. Discussion

This systematic review aimed to collect articles related to the relationship between overload and CBL. Only seven clinical studies were included; four showed a correlation between overload and CBL [92,96,97,98]. The main problem in this field of investigation is the proper assessment of the presence of overload and the quantification of its value; indeed, it is clinically difficult to measure occlusal forces during natural functioning [60]. Several methods have been proposed to detect excessive force on implant-supported fixed dental prostheses (FDPs), such as occlusal wear or wear facets, bruxism, reported tooth clenching, maximum bite force, the type of antagonist, and the length of the cantilever [52]. However, none of these signs are pathognomonic of the presence of overload, and they cannot provide a reproducible, quantifiable, absolute, or relative value. 

Lindquist et al. [92] evaluated factors related to CBL in 276 dental implants in 46 patients, divided in two groups by follow-up time (Group 1: 3–4 years; group 2: 5–6 years). The authors found that in group 1, the mean bone loss was positively correlated (*p* < 0.05) to the length of the cantilever extension. In group 2, the correlations were similar, but they did not reach a statistically significant level. After six years, seven patients with long cantilevers (length = 15 mm) had a mean loss of 0.95 mm around the mesial implant, whereas six patients with short cantilevers (length < 15 mm) had a loss of 0.61 mm. The presence of load in long cantilevers seems to generate higher forces on the mesial implant than on the posterior ones, like a lever creating tensile forces on medial implant. However, subsequent studies did not find any correlation between CBL and short cantilevers [93,94] or long cantilevers [15,91]. In group 2, clenching and recorded occlusal wear were found to be correlated with CBL with Spearman’s rank correlation coefficients of 0.41 (0.01 < *p* < 0.05) and 0.46, respectively (0.01 < *p* < 0.05). The same sample of patients was re-evaluated [93] for 12 to 15 years. According to the new statistical analysis, the factors related to CBL were poor oral hygiene, smoking, and anterior position of implants. Contrary to previous analysis, occlusal wear, the length of the cantilever, and reported tooth clenching did not present any significant correlation with CBL in the long-term follow-up. Carlsson et al. [94] conducted a cohort clinical study, with a follow-up of 15 years, on 47 edentulous patients rehabilitated with a mandibular complete-arch implant-supported FDP. During clinical examinations, 13 of the 47 patients received a maxillary complete-arch implant-supported FDP, whereas the other 34 had a removable complete denture as the antagonist. This study failed to find any correlation between occlusal wear and CBL. Similar results regarding the CBL in lower implants were recorded in patients with a complete-arch implant-supported FDP as the antagonist, as well as in those with a complete maxillary denture. This finding is in conflict with the observations of Quirynen et al. [15] and Naert et al. [88], who found a relation between CBL and the presence of an implant-supported FDP as the antagonist. The three studies discussed above were performed by one group of researchers, who followed the same selection criteria, treatment principles, and examination procedures [92,93,94]. The authors reported 11 implant losses in a total of 619 inserted implants, with 9 of the losses occurring before loading. The mean CBL was 0.9 mm both in the first sample of patients [92,93,94] and in the second one [94]. 

In the study of Engel et al. [95], all implant-supported FDPs presenting a shiny flat area or a flattening of the cusp tips were reported as wear facets. In this longitudinal study that evaluated implant-supported FDPs and overdentures on 379 patients, no correlation between occlusal wear and CBL was observed. However, occlusal wear was rare in implant-supported FDPs (14%) and more common in overdentures (43%) [99,100,101]. Among the studies included in the present review, only Engel et al. [95] investigated the influence of overload on peri-implant bone levels in the presence of different types of prostheses (i.e., overdentures, fixed partial prostheses, and single crowns), reporting no significant correlation. 

In a cross-sectional study of 938 implants with different follow-up times (1 to 14 years, mean = 5.64 years), Dalago et al. [96] found a prevalence of 7.3% for peri-implantitis. In the univariate analysis, wear facets on the prosthetic crown were positively associated with peri-implantitis, and the same correlation was demonstrated in the multi-factor analysis with OR [95% CI] = 2.4 [1.2–4.8] (*p* = 0.032). Similar findings in another cross-sectional study on 642 implants were reported by Kissa et al. [97]. In the univariate analysis in patients with occlusal wear on the posterior teeth, the mean probing depth (PD) of the implants was 5.69 mm and mean implant CBL was 2.26 mm, while in patients without posterior occlusal wear, these values were 4.77 mm (*p* = 0.004) and 1.85 mm (*p* = 0.02), respectively. In the multivariate analysis, the mean PD in patients with posterior occlusal wear was 5.94, while this value was 4.96 mm in patients without posterior occlusal wear (*p* = 0.01). In the multivariate analysis, mean CBL and wear facets did not reach a significant correlation (*p* = 0.2); however, a trend of higher CBL was observed in patients with wear facets (mean = 2.91 mm) than in those without wear facets (mean = 2.69 mm). Thus, the wear of a prosthesis can be considered a sign of occlusal dysfunction, and may be associated with the presence of overload. However, the identification of wear facets does not allow for the measurement of the magnitude and direction of the forces. 

Digital technologies are opening new possibilities for occlusal contact registration. In the retrospective case-control study carried out by Canullo et al. [98], static and dynamic contacts were recorded using a digital occlusal analysis device. The authors included only patients with at least two implants, with one or more affected by peri-implantitis. They assigned the implants affected by peri-implantitis (125) to the case group and the healthy implants (207) to the control group. Overload was reported in 3 implants in the healthy group and in 27 implants in the peri-implantitis group, with OR [95% CI] = 18.70 [5.5–63.2] (*p* = 0.0001). Therefore, excessive forces in an unbalanced occlusion were identified as the predictor of peri-implantitis. However, an accuracy of 82.35% and the small sample size were indicated as the limitations of the study by the authors. The combination of inadequate oral hygiene and signs of elevated occlusal load was found to have an impact on bone resorption [92], as well as on probing depth [97]. Moreover, other tools, such as the use of photoactivated blue-O toluidine [102], could be helpful in identifying peri-implantitis and preventing the development of the disease.

These findings are in agreement with those reported in other reviews [52,55,62] and in an animal study in monkey mandibles [37]. 

Contradictory results on the effects of overload on dental implants have also been found among animal studies. However, it has been frequently reported that overloaded implants exhibit higher bone–implant contact than unloaded implants [33,46]. This can be investigated only in animal models, owing to the possibility of sacrificing the animals and performing histological assessments of the peri-implant tissues. Therefore, load on implants seems to favor the quality of the peri-implant bone and enhance the osseointegration up to certain limits of load [103]. Above these values, the positive bone remodeling may involve a loss of osseointegration, mostly in presence of high lateral and dynamic forces [35,43]. Finally, in an animal model, peri-implant bone subjected to overload exhibited characteristic histological features. Unlike the implants affected by plaque-induced peri-implantitis, overloaded implants, which were losing osteointegration, presented a fibrous tissue between the bone and the implant surface, with a negligible inflammatory infiltrate in peri-implant soft tissues [49]. 

In summary, the main limitation encountered in the majority of the included studies, and in this field of research in general, arises from the utilization of assessment methods that are neither repeatable nor quantifiable. The absence of universally recognized threshold force values for the definition of overload is also a limitation for research studies on this topic. In the only included study that utilized a digital tool for assessing occlusal loading [98], overload was defined as an intensive red point resulting from the digital occlusal analysis system. The absence of a threshold value to use as a reference for digital occlusal checks in future research constitutes a limitation of the present review. Digital tools might represent a valid solution to overcome the limitations of traditional methods for the registration of occlusal contacts and the collection of quantitative data. These devices allow for the recording of occlusal forces, and for the intensity and presence of overload to be identified. Moreover, it is possible to convert the occlusal load into a numeric value that can be compared and analyzed. However, such tools are expensive. Conventional methods, such as articulating papers, impression waxes, and shim-stock foils, might be considered less reliable and objective as compared to digital tools [103]. Other limitations can be attributed to the restricted use of keywords during the electronic search strategy, which was complemented by an extensive manual search. However, the inclusion criteria would still have limited the results obtained. Finally, all the included studies presented a moderate to serious overall risk of bias.

## 5. Conclusions

Large clinical trials with long term follow-ups, which use repeatable and quantifiable assessment methods, are needed to clarify the role of overload on peri-implant bone loss. There is a need to identify a threshold value of overload that is able to trigger peri-implant bone loss. The use of standardized parameters would also allow for comparison among different clinical studies. The influence of implant features (e.g., design, connection, surface topography, diameter), restorative materials, and the type of implant-supported restorations in the onset and progression of overload-triggered peri-implant bone loss should be further explored. In this context, finite element analysis (FEA) may also contribute to a better definition of the influence of loading variation. For instance, FEA has been applied to investigate the influence on stress of the prosthetic designs associated with implants of varying lengths and distribution [104], of the prosthetic screw design [105], or of the distribution of occlusal contacts [106]. Another area that should be investigated concerns the extent to which the resolution of occlusal overload could be effective in limiting the progression of prosthetically triggered peri-implant bone loss. In the present review, the reported data relating the occlusal analysis to peri-implant bone level seem to reveal an association; this association must be further investigated using new digital tools that can help to standardize the methodology.

## Figures and Tables

**Figure 1 jcm-11-04812-f001:**
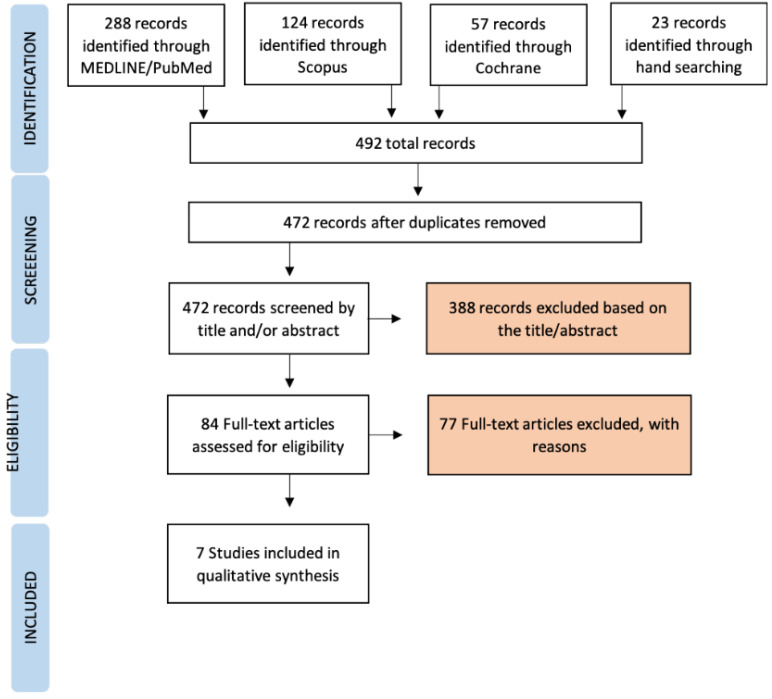
PRISMA flow chart of articles screened, withdrawn, and included in the review process.

**Table 1 jcm-11-04812-t001:** Main features of the included studies.

Author(s), Year	Study Design	Total No. of Patients	Total No. of Implants	No. of Compromized Implants	Implant Diameter and Length [mm] ^§^	Other Implant Features *	Type of Prosthesis	Follow-up (Range or ±SD) [Years] °	Occlusal Analysis	Correlation Overload–Cretal Bone Loss (Y/N)
Canullo et al., 2016 [98]	Retrospective case-control study	56	332	125	D < 4: 26 D = 4: 279 D > 4: 27 L not reported	Peri-implantitis group: Rough (*n* = 85) Smooth (*n* = 40)	Healthy implant group: Screwed (*n* = 127) Cemented (*n* = 80) Peri-implantitis group: Screwed (*n* = 64) Cemented (*n* = 61)	Healthy implant group: 6.48 ± 3.57 Peri-implantitis group: 5.94 ± 3.16	Fracture or chipping of the veneering; loss of retention; dynamic occlusal measurement by T-Scan III; occlusal photographs	Y **
Carlsson et al., 2000 [94]	Prospective cohort study	47 (of which 13 received treatment in both jaws)	343 (273 mandible; 75 maxilla)	8 (7 before loading)	D not reported L = 10 mm	Standard Brånemark implants (Nobel Biocare)	Full-arch implant-supported FDP (resin teeth)	15 (mandibular implants) 10.5 (8 to 13; maxillary implants)	Occlusal wear; bite force	N
Dalago et al., 2017 [96]	Retrospective cross-sectional study	183	938	89 (16 lost; 6 inactivated; 67 peri-implantitis)	D < 3.5 (*n* = 148) D = 3.5 (*n* = 575) D > 3.5 (*n* = 193) L < 9 (*n* = 796) L ≥ 9 (*n* = 120)	Connection: External Hexagon (*n* = 400) Internal Hexagon (*n* = 516)	Fixed restoration: Screwed (*n* = 436) Cemented (*n* = 480) Type of prosthesis: Single (*n* = 167) Partial (*n* = 522) Total (*n* = 227)	Mean: 5.64 (range 1 to 14)	Coronal fracture; wear facets	Y **
Engel et al., 2001 [95]	Prospective cohort study	379	379	21	D = 3.5 (*n* = 44) D = 3.5–4 (*n* = 153) D = 4.5–7 (*n* = 182) L not reported	Frialit-2 (*n* = 227) Bonefit (*n* = 51) IMZ (*n* = 47) Tübingen (*n* = 47) Brånemark (*n* = 6) TPS (*n* = 1)	Type of prostheses: Single (*n* = 188) Partial (*n* = 84) Overdenture (*n* = 107) Occlusal material: Ceramic (*n* = 182) Non-ceramic (*n* = 197)	Mean: 6 (range 1 to 10)	Wear facets	N
Lindquist et al., 1988 [92]	Prospective cohort study	46 25 (group 1) 21 (group 2)	276	N/A	Not reported	Brånemark implants	Mandibular full-arch implant-supported FDP	Group 1: 5½ to 6 Group 2: 3 to 4	Bite force; attrition and occlusal wear	Y
Lindquist et al., 1996 [93]	Prospective cohort study	47 26 (group 1) 21 (group 2)	273	3	Not reported	Brånemark implants	Mandibular full-arch implant-supported FDP Mean cantilever length left = 14.7 mm (7 to 20) Mean cantilever length right= 15 mm (7 to 20)	Group 1: 15 Group 2: 10	Bite force; attrition and occlusal wear	N
Kissa et al., 2020 [97]	Retrospective cross-sectional study	145	642	146	Not reported	SA (*n* = 221) SLA (*n* = 161) HA (*n* = 260)	Fixed restoration: Screwed (*n* = 436) Cemented (*n* = 480)	Mean: 6.4 (1 to 16) (1 to 3: 242) (4 to 8: 226) (>8: 174)	Occlusal wear	Y

^§^ No. of implants per type reported into brackets; * Roughness; connection etc.; ** correlation with peri-implantitis, which according to the provided definition includes CBL; ° based on implant age; D, implant diameter; FDP: fixed dental prosthesis; HA, hydroxylapatite particle-blasted and acid-washed surface; L: implant length; SA, sandblasted and acid-etched surface; SLA, sandblasted with large grit and acid-etched surface; TPS, titanium plasma-spray.

**Table 2 jcm-11-04812-t002:** Risk of bias assessment (ROBINS-I) L = “low risk of bias”; M = “moderate risk of bias”; S = “serious risk of bias”; C = “critical risk of bias”.

Study	Pre-Intervention		At Intervention	Post-Intervention				Overall Risk of Bias
	Confounding	Selection	Classification of Intervention	Deviation from Intended Intervention	Missing Data	Measurement of Outcome	Reporting Result	
Canullo et al., 2016 [98]	L	M	L	N/A	L	L	L	M
Carlsson et al., 2000 [94]	L	L	M	N/A	L	M	S	S
Dalago et al., 2017 [96]	L	M	M	N/A	L	L	L	M
Engel et al., 2001 [95]	L	L	M	N/A	S	M	M	S
Lindquist et al., 1988 [92]	L	L	M	N/A	L	M	S	S

N/A: not applicable.

## Data Availability

Not applicable.

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
