# Peer review of "Peri-Implant Bone Loss and Overload: A Systematic Review Focusing on Occlusal Analysis through Digital and Analogic Methods"

_jcm, 2022, doi:10.3390/jcm11164812_

Round 1

Reviewer 1 Report

Dear Authors, this topic is always very interesting and still remain as you have searched unclear, however your digital approach suggestion to measure in these dentistry era might give in the near future accurate results.

In your review you suggested in the Materials and Methods as transcribed in PICO question focused on digital assessment, how did you manage to find these approach as in the search strategy I do not see in the Search strategy this topic...

The search strategy applied was: 93 (dental or oral) AND implants AND (overload OR excessive load OR occlusal wear) AND 94 (bone loss OR peri-implantitis OR failure).

Have you conducted it manually after the results you got with the initial search?

In Discussion you state:

The absence of universally 285 recognized threshold force values for the definition of the overload is also a limitation for 286 research studies on this topic.

It would be interesting for the information you provide trough your systematic  review if you finded an occlusal threshold already described to write it on your article as other authors might use this as reference or useful to begin a digital occlusal check in future research.

Check words spelling...

Today, with the development of digital technology, it is possible to converter (192)

restricted use of kay-words during the search strategy, however, the inclusion criteria (288)

and individuate the presence of overload. (292)

Author Response

Dear Review,

Thank you so much for your suggestions and revisions. In the following attached file, We try to reply to all questions. 

Reviewer 2 Report

1.      Keywords need to be used in lowercase in the initial letter.

2.      The authors need to clarify the novelty of the present review. From the reviewer evaluation, there is nothing really new brings in this paper. Peri-implant bone loss has been widely explained in many reviews article. Highlighting this issue in the introduction section is mandatory. Present explanation by the authors is not enough.

3.      In lines 36-37, the sentence “Implant therapy represents a safe and predictable procedure to rehabilitate both 36 complete and partially edentulous patients” needs an additional reference to support this explanation. Adopted literature published by MDPI is strongly encouraged as follows: Computational Contact Pressure Prediction of CoCrMo, SS 316L and Ti6Al4V Femoral Head against UHMWPE Acetabular Cup under Gait Cycle. J. Funct. Biomater. 2022, 13, 64. https://doi.org/10.3390/jfb13020064

4.      Table 1. Is not important to show, delete it.

5.      Limitation of the present review needs to be included.

6.      The conclusion is too short. More elaboration is needed.

7.      Further research needs to be explained in the conclusion section.

8.      Please make sure the author has been following PRISMA 2020 properly.

9.      Recheck the MDPI template used for appropriate use. It needs to be revised due to several errors such as those that have been mentioned in previous comments.

Author Response

Dear Review, 

Thank you so much for your revisions and suggestions. In the following attached file, we try to reply to all questions. 

Reviewer 3 Report

In this systematic review the authors analyze the potential relationship between occlusal overload and peri-implant bone loss. Although an interesting subject, the authors must first address several issues before publication.

Please see enclosed pdf for further details.

Author Response

Dear Review, 

We apologize, but the referee has uploaded the comments to another publication and not to ours (Cárcamo-España et al., Extract or preserve from the periodontal point of view. Resolution of practical cases and review of the literature).

Round 2

Reviewer 2 Report

Thanks to the authors for their effort in revising their manuscript after peer review. I am read the revised version along with the response. I am not satisfied with the revised version, nothing significant improvement. Authors also fail to show something really new in the revised form since a peri-implant bone loss has been widely explained in the previous review literature. Scientific writing by the author is also messy, and unscientific, such as the example in the abstract which is inappropriate because it is too detailed and should not be necessary. Discussions related to digital and analogical do not explain anything meaningful. 

Author Response

Thank you so much for your revision, however, I don’t understand the suggestions of this last review. In the previous response, We try to answer all the requests ( I attached the Q&A). Please, Can you highlight the answers that did not satisfy? It’s true that the peri-implant bone has been widely investigated, however, there are not many articles that explain the relationship between overload and crestal bone loss. This review explained that digital tools are needed to record the overload. Second, the older articles seem to find a relationship, however, the clinicians before giving a diagnosis of overload should use digital tools and not only the clinical signs. I think that this message is fundamental for clinicians and researches

Regarding the discussion section, in the following sentences tried to explain the advantage and disadvantage of digital tools : “In summary, the main limitation encountered in the majority of the included studies, and in this field of research in general, consists of the utilization of not repeatable and quantifiable assessment methods. The absence of universally recognized threshold force values for the definition of the overload is also a limitation for research studies on this topic. In the only included study utilizing a digital tool for assessing occlusal loading [101], overload was defined as an intensive red point resulting from the digital occlusal analysis system. The absence of a threshold value to use as a reference for digital occlusal check in future researches constitutes a limitation of the present review. Digital tools might represent a valid solution to overcome the limitations of traditional methods for the registration of occlusal contacts and the collection of quantitative data. These devices allow to record the occlusal forces, identify the intensity, and detect the presence of overload. Moreover, it is possible to converter the occlusal load into a numeric value that can be compared and analyzed. However, the cost of these tools is expensive. Conventional methods, such as articulating papers, impression waxes, and shim-stock foils, might be considered less reliable and objective as compared to digital tools [106].

Reviewer 3 Report

I apologize for the confusion, I reincluded the PDF with the comments.

Author Response

Thank you so much for your suggestions. In the following sentences, we try to respond and to resolve the questions. 

Q:The abstract must be rewritten, too many details regarding methodology, and too few regarding the actual results

A: The abstract has been rewritten following the suggestions.

Q: The authors should stress the originality and the utility of this review in light of the overall literature.

A: The following sentences have been entered in the manuscript. “Overload potentially plays a role on the behavior of peri-implant bone, however its role on the onset and progression of bone loss is still unclear, therefore, a systematic review is needed to have a clear idea of the trouble”

Q: The discussions section is unstructured. Please reformulate and rearrange for better comprehension.

A: The discussion section has been modified, especially, in the first sentences. We try to give the following logic-flow:  Main results, discussion and compare the seven articles selected for this review, limitations, and possible future scenarios.

Q:The authors should discuss the treatment of peri-implantitis, including adjuvant treatments. I suggest:

Nicolae, V.; Chiscop, I.; Cioranu, V.S.I.; Martu, M.A.; Luchian, A.I.; Martu, S.; Solomon, S.M. The use of photoactivated blue-o toluidine for periimplantitis treatment in patients with periodontal disease. Rev. Chim. (Buchar.) 2015, 66, 2121–2123.

A: The following sentences and reference have been entered in the manuscript. “ Moreover, other tools such as the use of photoactivated blue-O toluidine [105] could be helpful in identifying peri-implantitis and preventing the development of the disease.”

Q:key-words, not kay-words

A:The word has been corrected.

Round 3

Reviewer 2 Report

Thanks for the author's explanation and their revision. I agree with the author regarding the relationship between overload and crestal bone loss have not many discussed in the literature. To extend the discussion, I am recommending the authors for discussing the potential study using the finite element analysis as a powerful tool for conducting in silico study. Suggested reference published by MDPI needs to be adopted as follows: Computational Contact Pressure Prediction of CoCrMo, SS 316L and Ti6Al4V Femoral Head against UHMWPE Acetabular Cup under Gait Cycle. J. Funct. Biomater. 2022, 13, 64. https://doi.org/10.3390/jfb13020064a

In the revised form, the authors construct a paragraph that only consists of two sentences see line 70-74, making it at least three sentences for a more constructive and clear explanation.

The conclusion section is missing, it needs to be added.

And the last, the reference needs to be enriched from literature published five years back. References published by MDPI is strongly recommended.

Author Response

Thanks for the author's explanation and their revision. I agree with the author regarding the relationship between overload and crestal bone loss have not many discussed in the literature.

A: Thank you for the positive feedback.

To extend the discussion, I am recommending the authors for discussing the potential study using the finite element analysis as a powerful tool for conducting in silico study. Suggested reference published by MDPI needs to be adopted as follows: Computational Contact Pressure Prediction of CoCrMo, SS 316L and Ti6Al4V Femoral Head against UHMWPE Acetabular Cup under Gait Cycle. J. Funct. Biomater. 2022, 13, 64. https://doi.org/10.3390/jfb13020064a.

A: We thank the referee for highlighting the importance to introduce the potential use of FEA in this area of research. We have accurately read the suggested paper (doi.org/10.3390/jfb13020064a), but unfortunately it refers to a completely different anatomical area and to a different kind of loading pattern. Indeed, the keywords related to the aforementioned paper are: “CoCrMo; SS 316L; Ti6Al4V; UHMWPE; contact pressure; total hip arthroplasty”. To support the potent use of FEA in this contest, three references (two from MDPI journals) have been added, i.e. Cenkoglu et al, 2019 [107], Farronato et al, 2019 [108], and Brune et al, 2019 [109].

In the revised form, the authors construct a paragraph that only consists of two sentences see line 70-74, making it at least three sentences for a more constructive and clear explanation.

A: Thank you for the suggestion. The whole paragraph has been revised and details have been added about animal studies.

The conclusion section is missing, it needs to be added.

A: Thank you for this observation. The last paragraph of the discussion constituted the conclusions, that are now clearly identifiable thanks to the introduction of the related subhead “5. Conclusions”. Moreover, the conclusion section has been further developed.

And the last, the reference needs to be enriched from literature published five years back. References published by MDPI is strongly recommended.

A: The following recently published MDPI references have been added:

- Farronato D, Manfredini M, Stevanello A, Campana V, Azzi L, Farronato M. A Comparative 3D Finite Element Computational Study of Three Connections. Materials (Basel). 2019;12(19):3135. doi: 10.3390/ma12193135.

- Cenkoglu BG, Balcioglu NB, Ozdemir T, Mijiritsky E. The Effect of the Length and Distribution of Implants for Fixed Prosthetic Reconstructions in the Atrophic Posterior Maxilla: A Finite Element Analysis. Materials (Basel). 2019 Aug 11;12(16):2556. doi: 10.3390/ma12162556.

Reviewer 3 Report

The manuscript has been improved

Author Response

Thank you for the positive feedback.